# Effect of *Hermetia illucens* Fat, Compared with That of Soybean Oil and Palm Oil, on Hepatic Lipid Metabolism and Plasma Metabolome in Healthy Rats

**DOI:** 10.3390/ani13213356

**Published:** 2023-10-29

**Authors:** Robert Ringseis, Magdalena J. M. Marschall, Sarah M. Grundmann, Sven Schuchardt, Erika Most, Denise K. Gessner, Gaiping Wen, Klaus Eder

**Affiliations:** 1Institute of Animal Nutrition and Nutrition Physiology, Justus Liebig University Giessen, Heinrich-Buff-Ring 26-32, 35392 Giessen, Germany; robert.ringseis@ernaehrung.uni-giessen.de (R.R.); magdalena.marschall@ernaehrung.uni-giessen.de (M.J.M.M.); sarah.grundmann@ernaehrung.uni-giessen.de (S.M.G.); erika.most@ernahrung.uni-giessen.de (E.M.); denise.gessner@ernaehrung.uni-giessen.de (D.K.G.); gaiping.wen@ernaehrung.uni-giessen.de (G.W.); 2Fraunhofer Institute for Toxicology and Experimental Medicine (ITEM), Hannover, Nikolai-Fuchs-Str.1, 30625 Hannover, Germany; sven.schuchardt@item.fraunhofer.de; 3Center for Sustainable Food Systems, Justus Liebig University Giessen, Senkenbergstraße 3, 35390 Giessen, Germany

**Keywords:** insect fat, *Hermetia illucens*, soybean oil, palm oil, hepatic lipid metabolism, plasma metabolome, healthy rats

## Abstract

**Simple Summary:**

Palm oil is currently the most widely used fat source for food production, but palm oil production is associated with severe environmental problems. Insect fat from *Hermetia illucens* larvae might be a suitable alternative fat source, because its production is more sustainable and less harmful to the environment. Thus, the present study investigated the effect of *Hermetia* fat, as compared to palm oil and soybean oil, on the hepatic lipid metabolism and the plasma metabolome of healthy rats, which were fed diets containing either soybean oil, palm oil, or *Hermetia* fat for 4 weeks. Growth performance, liver and plasma lipid concentrations, and the expression of hepatic genes involved in lipid metabolism and inflammation did not differ between groups. Plasma metabolomics revealed a clear separation of the plasma metabolomes of the soybean oil group and the other two groups, but not of those of the palm oil and the *Hermetia* fat group. The present study shows that *Hermetia* fat exerts no adverse effects on lipid metabolism and inflammatory gene expression in the liver of healthy rats compared to palm oil or soybean oil. Thus, the present findings indicate that *Hermetia* fat is a safe alternative fat source to palm oil for food production.

**Abstract:**

Palm oil (PO) is currently the most widely used fat source for food production, but insect fat from *Hermetia illucens* larvae (HF) might be a suitable alternative fat source, because its production is less harmful to the environment. The present study investigated the effect of HF, as compared to PO and soybean oil (SO), on the hepatic lipid metabolism and the plasma metabolome of healthy rats, which were randomly assigned to three groups (*n* = 10 rats/group), and fed three different semi-synthetic diets containing either SO, PO, or HF as the main fat source for 4 weeks. Feed intake, body weight gain, liver and plasma lipid concentrations, and the hepatic mRNA levels of genes involved in lipid metabolism and inflammation did not differ between groups. Targeted plasma metabolomics revealed 294 out of 630 metabolites analyzed to be different between groups. Principal component analysis showed a clear separation of the plasma metabolomes of the SO group and the other two groups, but no separation of those of the PO and the HF groups. The present study shows that HF exerts no adverse metabolic effects in healthy rats, compared to PO or SO, indicating that HF is a safe alternative fat source to PO for food production.

## 1. Introduction

Owing to its semisolid consistency at room temperature, palm oil (PO) is well suited to the production of processed foods, like ready meals, instant soups, biscuits, cakes, spreads, or chocolate, thereby explaining its widespread use [1,2]. However, as the expansion of PO plantations in Asia and South America is associated with severe environmental problems, alternatives are needed to meet the growing global demand for fat for food production.

In recent years, insect fat obtained from the industrial farming of suitable insect larvae, e.g., *Hermetia illucens* larvae, has received great attention as a sustainable and environmentally friendly source of fat [3,4]. In fact, insect larvae can be reared on low-value/low-quality agricultural sidestreams, which are regionally available from local agro-industrial companies, thereby avoiding the greenhouse gas emissions resulting from long transport routes [5], which are necessary for imported plant oils or oil seeds. In addition, the production of insect biomass results in fewer greenhouse gas emissions and requires significantly less land when compared to other animal-derived products, like meat and milk [6]. Like PO, *Hermetia illucens* larvae fat (HF) contains mainly saturated fatty acids (SFAs), in particular the medium-chain fatty acid (MCFA) lauric acid (C12:0; 40–55% of total fatty acids [7,8]), which explains its similar consistency at room temperature to PO, and makes HF a suitable alternative fat source for food production, provided that HF does not cause any undesirable metabolic effects. Concerns in this regard may arise from the high SFA content in HF, because SFAs are widely considered to be detrimental to cardiovascular health, which explains why dietary guidelines for the prevention of cardiovascular disease comprise a decreased intake of SFA [9,10,11]. Thus, studies investigating the impact of HF on lipid metabolism compared to PO are of great relevance. However, studies in this regard are completely lacking.

In view of this, the aim of the present study was to investigate the effect of HF compared with PO on the hepatic lipid metabolism and the plasma metabolome of healthy rats. In addition, soybean oil (SO)—a fat rich in polyunsaturated fatty acids (PUFA)—was used as a further reference fat.

## 2. Materials and Methods

### 2.1. Animals and Diets

The animal experiment was approved by the Animal Welfare Officer of the Justus Liebig University Giessen (approval no.: JLU 790_M). All experimental procedures described followed established guidelines for the care and handling of laboratory animals. Thirty male (7–8 weeks of age) homozygous (fa/fa) lean Zucker rats (Crl:ZUC-Lepr^fa/+^) with an average body weight of 242 ± 29 g (mean ± SD, *n* = 30) served as experimental animals, which were obtained from Charles River (Sulzfeld, Germany). The rats were randomly assigned to three groups of 10 rats each and housed in groups of two animals each in a controlled environment with 12 h light/12 h dark cycle, 22 ± 1 °C ambient temperature, and 50–60% relative humidity. The three groups received three different semi-synthetic diets, which varied in the main fat source (SO, PO, HF; Table 1). Nutrient levels in the diets were sufficient to meet the requirements of the rat for growth according to the National Research Council (NRC) [12]. The SO diet contained SO as the sole fat source (70 g/kg diet). The PO diet and the HF diet contained 68.6 g/kg diet of PO and HF, respectively, and 0.7 g/kg diet of SO and 0.7 g/kg diet of linseed oil in order to cover the demand of C18:2 n-6 and C18:3 n-3. The SO and the PO were purchased from Chemiekontor (Mannheim, Germany). The HF was obtained from a local company (Madebymade, Pegau, Germany). The HF fat consisted mainly of triglycerides (TG) (>99%) as comprehensively analyzed recently using lipidomics [13]. The total lipid fatty acid composition of the experimental diets, which was determined via gas chromatography-flame ionization detection (GC-FID) [14], is also shown in Table 1. In order to determine the apparent ileal digestibility (AID) via the indicator method, all diets contained 0.5% titanium dioxide (TiO_2_). The experimental diets and water were provided to the rats ad libitum for a period of 4 weeks.

### 2.2. Sample Collection

Rats were sacrificed under CO_2_ anesthesia, and blood was collected in heparin-coated polyethylene tubes (AppliChem, Darmstadt, Germany). Blood was centrifuged (1100× *g*, 10 min, 4 °C) to obtain plasma. The liver was excised, washed in 0.9% NaCl solution, weighed and several aliquots were taken separately. The gut was removed and digesta collected from the cecum. In addition, the right kidney and three different muscles (*Soleus*, *Gastrocnemius*, *Rectus femoris*) from the right leg were excised and weighed. All tissue samples were snap-frozen in liquid nitrogen and stored at −80 °C pending analysis.

### 2.3. Determination of AID of Total Fat

At the end of the experiment the AID of total fat was measured via the indicator method using TiO_2_ [15,16]. Processing of ileal digesta samples and calculation the AID was carried out as described recently [13].

### 2.4. Determination of Total Lipid Concentration

Total lipid concentration of the ileum chyme and the feed was analyzed via a colorimetric method according to Zöllner and Kirsch [17]. In brief, freeze-dried feed (150 mg) and ileum chyme (100 mg) were homogenized with 1 mL n-hexane and isopropanol (3:2, *v/v*) in a TissueLyser (Qiagen, Hilden, Germany) at 30 Hz for 3 min. Afterwards, the homogenate was incubated in an ultrasonic bath for 30 min, and finally centrifuged (4500 rpm, 15 °C, 10 min). An aliquot of the supernatant was evaporated under a stream of N_2_ at 37 °C. The dried lipids were dissolved in 2 mL of concentrated H_2_SO_4_, and then heated at 105 °C for 10 min. After reaching room temperature, a 50 μL aliquot was added to 1 mL vanillin-phosphorus reagent (0.6 g vanillin reagent dissolved in 500 mL water and o-phosphoric acid (1:4, *v/v*)), and the mixture was incubated for 40 min at room temperature. Finally, the absorption of the solution was determined photometrically at 530 nm, and the concentration of total lipids was calculated using an external standard curve.

### 2.5. Histological Evaluation of Liver Lipid Accumulation

The accumulation of liver lipids was visualized via Oil Red O (ORO) staining and Haematoxylin and Eosin staining of cryosectioned liver slices using an EVOS M5000 microscope (Thermo Fisher Scientific, Dreieich, Germany).

### 2.6. Determination of TG and Cholesterol Concentrations in Plasma and Liver

Concentrations of TG and cholesterol (Chol) in plasma and total lipid extracts from the liver were measured using commercial kits from Analyticon Biotechnologies (Lichtenfels, Germany). Liver total lipids were extracted using a mixture of n-hexane and isopronanol (3:2, *v/v*) [18], and extracted lipids were dried and dissolved with chloroform and Triton X-100 (1:1, *v/v*) [19].

### 2.7. Total RNA Isolation and qPCR Analysis

Total RNA extraction of liver aliquots, cDNA synthesis, and qPCR analysis using gene-specific primers (Appendix A) was performed as recently described [8]. The qPCR data were normalized using the three most stable (*Canx*, *Mdh1*, *Rpl13*, *Sdha*) out of seven potential reference genes tested, according to [20].

### 2.8. Targeted Plasma Metabolomics

Quantification of targeted plasma metabolites was carried out using a combination of liquid chromatography (Agilent 1290 Infinity II LC, Santa Clara, CA, USA) and mass spectrometry (SCIEX 5500 QTrap™ MS, Darmstadt, Germany) using the MxP™ Quant 500 kit (BIOCRATES Life Sciences AG, Innsbruck, Austria) as described recently in detail [21]. All metabolomics data were analyzed using the MetaboAnalystR 3.2 package for R version 4.2.1 [22].

### 2.9. Statistical Analysis

All data were analyzed using SPSS 27 statistical software (IBM, Armonk, New York, NY, USA). The individual animal served as the experimental unit for all data, with the exception of daily feed intake (the cage served as the experimental unit). Distribution of normality was assessed via Shapiro–Wilk test. Homogeneity of variance was evaluated using Levene´s test. In the case of normally distributed and variance homogeneous data, data were analyzed via one-way ANOVA followed by a Tukey’s post hoc test. In the case of variance heterogenous data, the means of the three groups were analyzed using Welch’s ANOVA combined with a Games-Howell post hoc test. Data which were not normally distributed were analyzed using Mann–Whitney U test coupled with Bonferroni correction. A *p*-value < 0.05 was considered statistically significant. The metabolomics dataset was subjected to principal component analysis (PCA) and the individual metabolites were subjected to statistical single factor comparisons with Tukey’s post hoc test. Metabolites with a false discovery rate (FDR)-adjusted *p*-value < 0.05 were considered significantly different. Variables with absent values were either excluded from analyses in the case that >50% of samples were missing, or missing values were substituted by the limit of detection. The remaining values were used for analysis after normalization.

## 3. Results

### 3.1. Body Weight Development, Feed Intake, Organ Weights, and AID of Total Fat

Final body weight, daily body weight gain, and daily feed intake did not differ among groups (Table 2). Liver and kidney weights and the weights of selected skeletal muscles were not different among groups. The AID of total fat was lower in the HF group than in the SO group (*p* < 0.05), but did not differ between the HF group and the PO group.

### 3.2. Lipid Concentrations in Liver and Plasma

Concentrations of TG and Chol in the liver did not differ among the groups (Figure 1A). The ORO and H&E staining of liver cryosections revealed the normal appearance of the parenchyma structure with normal liver cell morphology, clear edges, clearly visible nuclei, and no pathological lipid accumulation in all groups (Figure 1B). While the plasma concentration of TG tended to be higher in the HF and the PO groups than in the SO group (*p* < 0.10), the plasma Chol concentration was not different among the groups (Figure 1C).

### 3.3. Fatty Acid Concentrations in Liver Total Lipids

While the concentrations of total fatty acids did not differ among groups, there were differences between the groups in the concentrations of several individual fatty acids (Table 3). The concentrations of C12:0, C14:0, and C16:1 n-7, whose levels were highest in the HF fat, were higher in the HF group than in the SO group and the PO group (*p* < 0.05) indicating that liver fatty acid composition reflected the dietary fatty acid composition. The concentrations of these fatty acids did not differ between the latter two groups. The concentrations of C18:1 n-9 and C18:3 n-3 were higher in the HF group and the PO group than in the SO group (*p* < 0.05). The concentrations of C18:2 n-6, C18:3 n-6, C20:2 n-6, and C22:5 n-3 were lower in the HF group than in the SO group, but the concentrations of C18:2 n-6 and C18:3 n-6 were higher in the HF group than in the PO group (*p* < 0.05). The concentrations of C14:1 n-5, C16:0, C18:0, C20:3 n-6, C20:4 n-6, and C22:6 n-3 in the liver were not different among the groups.

### 3.4. Fatty Acid Concentrations in Skeletal Muscle Total Lipids

Like in the liver, the concentrations of total fatty acids were not different across the three groups (Table 4). The concentration of C12:0, the dominating fatty acid in HF, was higher in both the HF group and the PO group than in the SO group (*p* < 0.05), but did not differ between the HF group and the PO group. The concentrations of C18:2 n-6 and C18:3 n-3 were lower in the HF and PO groups compared to the SO group (*p* < 0.05). The concentrations of C22:5 n-3 and C22:6 n-3 were lower in the PO group than in the SO group (*p* < 0.05), but were not different between the HF group and the SO group, and the HF group and the PO group. The concentrations of all other fatty acids detected (C14:0, C14:1 n-5, C16:0, C16:1 n-7, C17:0, C18:0, C18:1 n-9) were not different across the three groups.

### 3.5. Hepatic Expression of Genes Involved in Lipid Metabolism and Inflammation

The hepatic mRNA levels of genes involved in fatty acid and TG synthesis (*Acaca*, *Acly*, *G6pd*, *Gpam*), fatty acid elongation and desaturation (*Elovl5*, *Scd1*), Chol homeostasis (*Hmgcr*, *Ldlr*), fatty acid oxidation (*Acox1*, *Cpt1a*), bile acid synthesis (*Cyp7a1*), and inflammation (*Crp*, *Icam1*, *Il1b*, *Tnfa*), did not differ among groups (Figure 2).

### 3.6. SCFA Concentrations in the Cecum Digesta

Concentrations of total (Figure 3A) and individual SCFA (acetate (C2), propionate (C3), isobutyrate (iC4), butyrate (C4), isovalerate (iC5), and valerate (C5); Figure 3B) in the cecum digesta were not different among groups.

### 3.7. Identification of Altered Plasma Metabolites Using Targeted Metabolomics

The concentrations of 294 out of the 630 metabolites analyzed were different across the three groups (FDR < 0.05). Most of these metabolites belonged to the class TG (206). The remaining metabolites were phosphatidylcholines (36), Chol esters (12), diglycerides (12), lysophosphatidylcholines (7), sphingomyelins (7), fatty acids (6), amino acids-related (2), ceramides (2), hexosylceramides (2), acylcarnitines (1), and indole derivatives (1). All altered metabolites are shown in the Appendix A. Out of the 294 metabolites which were different among the three groups, 243 differed between the SO group and the PO group, 257 differed between the SO group and the HF group, and 185 differed between the PO group and the HF group. The plasma metabolites which were different between the SO group and the PO group were TG (174), phosphatidylcholine (28), diglycerides (10), Chol esters (9), sphingomyelins (6), fatty acids (6), lysophosphatidylcholines (5), amino acids-related (2), ceramides (1; Cer(d18:1/24:1)), glycosylceramides (1; HexCer(d18:1/24:1)), indole derivatives (1; 3-IPA). The plasma metabolites which were different between the SO group and the HF group were TG (185), phosphatidylcholines (27), diglycerides (12), Chol esters (9), lysophosphatidylcholines (6), sphingomyelins (6), fatty acids (5), ceramides (2), glycosylceramides (2), acylcarnitines (1; C12), amino acids-related (1; betaine), and indole derivatives (1; 3-IPA). The plasma metabolites which were different between the PO group and the HF group were TG (135), phosphatidylcholines (27), Chol esters (8), lysophosphatidylcholines (4), diglycerides (3), fatty acids (3), sphingomyelins (3), acylcarnitines (1; C12), and amino acids-related (1; ornithine).

The dimensional reduction of the metabolome dataset via PCA analysis, which was carried out with all analyzed plasma metabolites, showed a clear separation (rightward shift) between the SO group and the other two groups (Figure 4A), whereas no clear separation was found between the PO group and the HF group. The cumulative proportion is 67.4%, with principal component 1 accounting for 41% of the variance of the dataset. The loading plot provides insight into the parameters that drive the separation of the data along the principal component. The red dots in the loading plot mark the ten parameters in each direction (top, bottom, left, and right) that contribute most to the separation of the data points along the principal component. The loading plot shows that the left shift of the PO group and the HF group was mainly caused by TG species with two or three double bonds, such as TG(16:0_38:3), TG(18:1_34:2), TG(16:0_32:3), TG(20:2_32:1), TG(16:1_34:2), TG(18:1_32:2), TG(14:0_34:2), TG(16:0_35:2), TG(16:0_32:2), and TG(16:1_36:2). In contrast, the right and upwards shift of the SO group was mainly caused by TG species with four and more double bonds, such as TG(18:2_34:2), TG(18:2_38:4), TG(18:2_34:2), TG(16:0_36:4), TG(18:2_35:2), TG(18:1_38:7), TG(18:3_34:2), TG(16:0_36:5), TG(18:3_34:3), TG(20:3_36:4), TG(20:4_36:4), TG(18:2_36:4), and TG(18:3_36:4) (Figure 4B).

## 4. Discussion

Considering the lack of studies investigating the metabolic effects of HF, which may be suitable as an alternative fat source to PO for food production, the present study evaluated the effect of HF, compared with PO and SO, on the hepatic lipid metabolism and the plasma metabolome of healthy rats.

The main finding of the present study is that HF exerts no liver and plasma TG- and Chol-modulating effects as compared to PO and SO in healthy rats. In line with this, no effect was found between the three groups of rats with regard to the hepatic expression of genes involved in lipid metabolism. Also, the hepatic expression of inflammatory genes did not differ between the three groups, indicating that HF does not induce any pro-inflammatory effect in the liver, which is known to impair hepatic function and to promote fatty liver development [23]. Collectively, these findings suggest that HF, compared to PO, has no undesirable effects on liver function and lipid metabolism in healthy rats. 

While the concentration of total fatty acids in the liver total lipids did not differ between groups, which is in accordance with the unaltered concentrations of hepatic TG among the three groups, pronounced differences were found in the concentrations of individual fatty acids in the liver total lipids between groups. However, it was noticeable that the differences in the concentrations of individual fatty acids in liver and skeletal muscle total lipids between groups were markedly stronger between the SO group and the other two groups (PO and HF) than between the HF group and the PO group. This was particularly evident for the concentrations of PUFA, like C18:2 n-6 and C18:3 n-3, which were found in markedly lower levels in the liver and skeletal muscle of the HF and PO rats than in the SO rats. Even the hepatic concentrations of C12:0 and C14:0, which occur in markedly higher levels in HF than in PO, were only slightly higher in the HF group than in the PO group. 

The more pronounced differences in the fatty acid composition of liver and skeletal muscle between the SO group and the other two groups (PO and HF), than between the HF and the PO group, were also reflected in the plasma metabolome analysis. Dimensional reduction of the metabolome dataset via PCA analysis revealed a clear separation of the plasma metabolomes of the SO group and the other two groups (PO and HF), whereas no separation between the plasma metabolomes of the PO group and the HF group was seen. According to the loading plot, the combined left shift of the metabolomes of the PO group and the HF group was mainly caused by TG species with two or three double bonds, whereas the right and upwards shift of the metabolome of the SO group was mainly caused by TG species with four and more double bonds. In addition, the more distinct separation of the plasma metabolome of the SO group from that of the other groups was evident from the total number of TG species differing between groups; while 185 and 174 TG species differed between the SO group and the HF group and between the SO group and the PO group, respectively, only 135 TG species were different between the PO and the HF group. In line with the high amount of C12:0 in HF fat compared to PO, free C12:0 and dodecanoylcarnitine were two plasma metabolites differing between the HF and the PO group.

We also analyzed the concentrations of SCFA in the cecal digesta of the rats. SCFA are the main microbial fermentation products in the gut, and it is well documented that altered concentrations of total and individual SCFA are typically caused by a shift in the gut bacterial community [24]. This is due to the fact that the metabolic pathways engaged in substrate utilization by different bacterial taxa markedly differ. Although we did not analyze the gut microbiome of the rats, the unaltered concentrations of total and individual SCFA in the cecal digesta of the rats across the three groups suggests that there was no profound effect of HF or the other fats. While the present study does not provide an answer to the question about whether the fat from *Hermetia illucens* modulates the gut microbiome of healthy rats, a large number of studies exist demonstrating the effect of either the fat or the larvae meal from *Hermetia illucens* on the gut microbiome and the intermediary metabolism of monogastric animals [11,25,26]. Thus, it is not unlikely that HF also modulates the gut microbiota of healthy rats.

Despite concerns existing with regard to a high intake of SFA, because SFA are considered to impair cardiovascular health, these concerns are likely unfounded in the case of HF. Unlike PO, which contains approximately 45% of its total fatty acids as long-chain SFA, the vast majority of SFA in HF are MCFA, in particular lauric acid. Based on clinical evidence from randomized controlled trials, fats containing high levels of MCFA, like coconut oil, are less detrimental to cardiovascular risk factors compared to other SFA-rich fats, like butter, which contain mainly long-chain SFA [27]. Despite the fact that no adverse effects of HF were found in the present study, firm conclusions about the safety of HF cannot be drawn from this study due to the rather short feeding period. In addition, the transfer of the data to the human context is limited by the fact that the fat content of the diets was comparatively low. While the rat diets of the present study contained 16% of their calories from fat, typical human diets in westernized countries contain 35-40% of their calories from fat.

## 5. Conclusions

The present study shows that HF exerts no adverse effects on lipid metabolism and inflammatory gene expression in the liver of healthy rats compared to PO or SO. Tissue fatty acid analysis and plasma metabolomics revealed only marginal differences between rats fed HF and PO, whereas pronounced differences were seen, as anticipated, between rats fed either HF or PO and rats fed SO. Thus, the present findings indicate that HF is a potential alternative fat source to PO for food production. Long-term feeding studies and studies with higher dietary fat levels are required to address the safety issues associated with HF intake.

## Figures and Tables

**Figure 1 animals-13-03356-f001:**
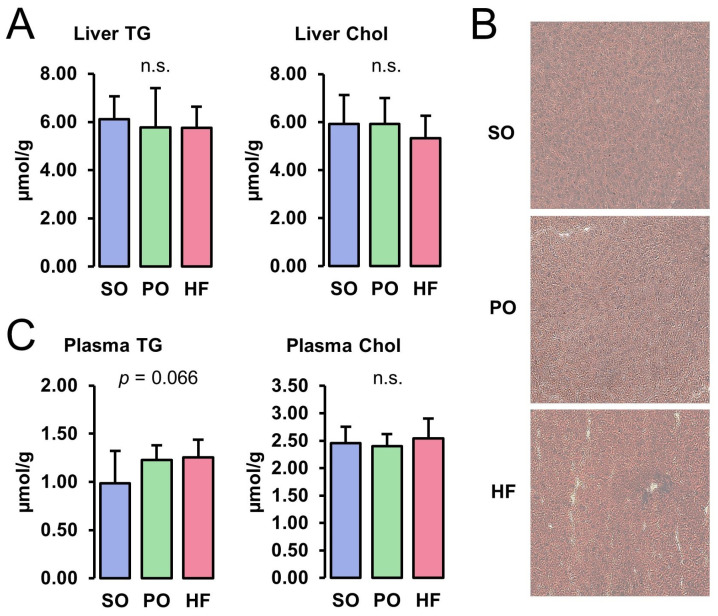
Liver and plasma lipid concentrations: Hepatic concentrations of TG and Chol (**A**), ORO-stained liver sections (**B**), and plasma concentrations of TG and Chol (**C**) of lean Zucker rats fed semi-synthetic diets with either soybean oil (SO), palm oil (PO), or *Hermetia* fat (HF) as the main fat source for 4 weeks. (**A**) Data are means ± SD for *n* = 10 rats/group. (**B**) Images are shown for one animal per group and are representative of all animals analyzed per group.

**Figure 2 animals-13-03356-f002:**
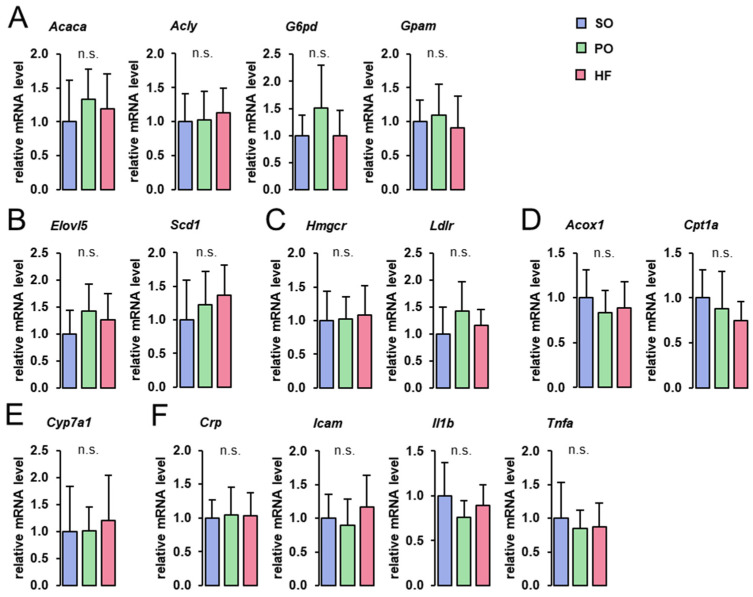
Hepatic expression of genes involved in lipid metabolism and inflammation: Hepatic mRNA concentrations of genes involved in fatty acid and TG synthesis (**A**), fatty acid elongation and desaturation (**B**), Chol homeostasis (**C**), fatty acid oxidation (**D**), bile acid synthesis (**E**), and inflammation (**F**) in the liver of lean Zucker rats fed semi-synthetic diets with either soybean oil (SO), palm oil (PO), or *Hermetia* fat (HF) as the main fat source for 4 weeks. Data are means ± SD for *n* = 10 rats/group.

**Figure 3 animals-13-03356-f003:**
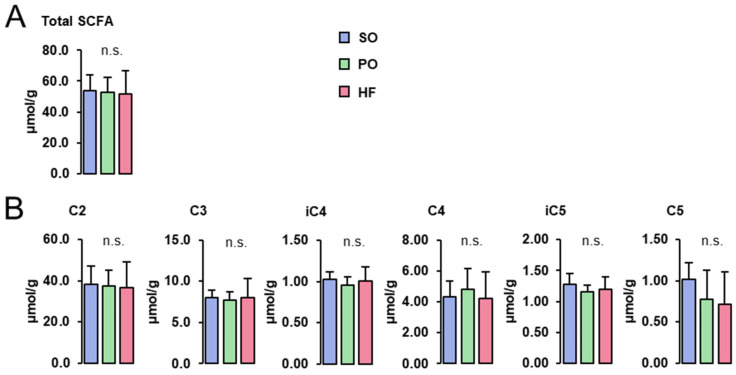
SCFA concentrations in the cecum digesta: Concentrations of total (**A**) and individual SCFA (**B**) in the cecum digesta of lean Zucker rats fed semi-synthetic diets with either soybean oil (SO), palm oil (PO) or *Hermetia* fat (HF) as the main fat source for 4 weeks. Data are means ± SD for *n* = 10 rats/group.

**Figure 4 animals-13-03356-f004:**
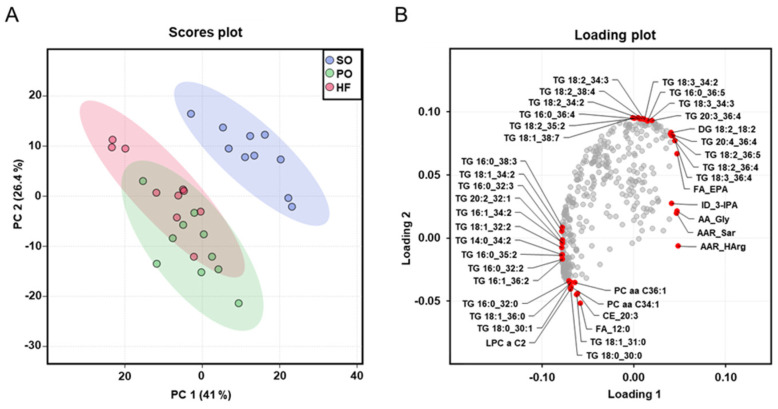
Principal component analysis (PCA) of the plasma metabolome: Scores plot with plotted 5% confidence interval (**A**) and associated loading plot (**B**) of PCA of the plasma metabolome of lean Zucker rats fed semi-synthetic diets with either soybean oil (SO), palm oil (PO), or *Hermetia* fat (HF) as the main fat source for 4 weeks. Data are principal components (PC 1 or PC 2) and their loadings for *n* = 10 rats/group. In the loading plot the ten parameters in each direction (top, bottom, left, and right) that contribute most to the separation of the data points along the PC are labeled in red. Abbreviations: AA, amino acid; AAR, amino acids-related; DG, diglyceride; FA, fatty acid; ID, indole; lysoPC, lysophosphatidylcholine; PC aa, phosphatidylcholine; PC ae, phosphatidylcholine ether; TG, triglyceride.

**Table 1 animals-13-03356-t001:** Diet composition and fatty acid composition of dietary total lipids.

	SO Diet	PO Diet	HF Diet
*Component, g/kg*			
Cornstarch	525.6	525.6	525.6
Casein	200	200	200
Sucrose	100	100	100
Mineral mix ^1^	35	35	35
Vitamin mix ^2^	10	10	10
Soybean oil	70	0.7	0.7
Palm oil	-	68.6	-
*Hermetia illucens* fat	-	-	68.6
Linseed oil	-	0.7	0.7
Cellulose	50	50	50
L-cysteine	4.4	4.4	4.4
TiO_2_	5	5	5
*Fatty acid composition of dietary total lipids, g/100 g total fatty acids*			
C10:0	0.01	0.03	1.1
C12:0	0.39	0.66	54.8
C14:0	0.22	1.19	8.78
C16:0	10.6	39.9	11
C16:1 n-7	0.11	0.21	2.64
C18:0	4.81	4.5	1.49
C18:1 n-9	23.1	39.9	7.65
C18:2 n-6	51.7	10.4	9.6
C18:3 n-6	0.2	0.49	0.29
C18:3 n-3	5.66	0.91	1.35
C20:0	1.21	0.59	0.36
C20:1 n-9	0.25	0.15	0.04
C20:4 n-6	0.26	0.43	0.14
C22:0	0.71	0.36	0.36
C22:1 n-9	0.57	0.1	0.23
SFA/UFA ratio	0.22	0.9	3.55
n-6 PUFA/n-3 PUFA ratio	9.22	12.44	7.43

^1^ The mineral mix provided the following per kg diet: calcium, 5 g; potassium, 3.6 g; chloride, 1.57 g; phosphorus, 1.56 g; sodium, 1.02 g; magnesium, 0.51 g; iron, 35 mg; zinc, 30 mg; manganese, 10 mg; copper, 6 mg; chromium, 1 mg; fluoride, 1 mg; iodate, 0.2 mg; molybdate, 0.15 mg; selenium; 0.15 mg; lithium, 0.10 mg. ^2^ The vitamin mix provided the following per kg diet: all-trans-retinol, 1.2 mg; cholecalciferol, 0.025 mg; menadione sodium bisulfate, 0.75 mg; all-rac-α tocopheryl acetate, 50 mg; thiamine HCl, 5 mg; riboflavin, 6 mg; pyridoxine HCl, 6 mg; cyanocobalamine, 0.025 mg; biotin, 0.2 mg; folic acid, 2.0 mg; nicotinic acid, 30 mg; pantothenic acid, 15 mg; choline, 1000 mg. Abbreviations: PUFA, polyunsaturated fatty acids; SFA, saturated fatty acids; UFA, unsaturated fatty acids.

**Table 2 animals-13-03356-t002:** Body weight development, feed intake, organ weights, and apparent ileal digestibility (AID) of total fat of lean Zucker rats fed semi-synthetic diets with either soybean oil (SO), palm oil (PO), or *Hermetia* fat (HF) as the main fat source for 4 weeks.

	SO	PO	HF	*p*-Value
Final body weight	343 ± 34	351 ± 26	351 ± 24	0.769
Daily body weight gain, g	3.64 ± 0.83	3.93 ± 0.55	3.81 ± 0.44	0.589
Daily feed intake, g	20.6 ± 1.8	22.2 ± 1.5	21.8 ± 0.8	0.239
Organ weights				
Liver, g	12.7 ± 2.3	13.0 ± 1.4	14.0 ± 1.3	0.210
Liver to body weight ratio, mg/g	3.67 ± 0.37	3.70 ± 0.21	4.00 ± 0.44	0.080
Kidney, g	1.16 ± 0.14	1.15 ± 0.06	1.21 ± 0.11	0.457
*M. soleus*, g	0.12 ± 0.02	0.13 ± 0.01	0.13 ± 0.01	0.595
*M. gastrocnemius*, g	1.77 ± 0.23	1.85 ± 0.17	1.89 ± 0.14	0.392
*M. rectus femoris*, g	0.91 ± 0.14	1.01 ± 0.13	0.99 ± 0.13	0.199
AID Total fat, %	95.2 ± 3.1 ^a^	93.0 ± 2.4 ^ab^	89.3 ± 4.1 ^b^	0.004

Data are means ± SD for *n* = 10 rats/group (body weight, daily body weight gain, organ weights, AID Total fat), *n* = 5 rats/group (daily feed intake). Means not sharing the same superscript letter (^a, b^) differ across groups, *p* < 0.05.

**Table 3 animals-13-03356-t003:** Fatty acid concentrations in the liver of lean Zucker rats fed semi-synthetic diets with either soybean oil (SO), palm oil (PO), or *Hermetia* fat (HF) as the main fat source for 4 weeks.

	SO	PO	HF	*p*-Value
*Fatty acid, µmol/g*				
C12:0	0.64 ± 0.17 ^b^	0.64 ± 0.15 ^b^	0.91 ± 0.18 ^a^	0.001
C14:0	0.91 ± 0.32 ^b^	1.11 ± 0.18 ^b^	1.62 ± 0.24 ^a^	<0.001
C14:1 n-5	0.42 ± 0.21	0.37 ± 0.07	n.d.	0.880
C16:0	27.4 ± 3.3	28.1 ± 2.4	29.7 ± 3.5	0.260
C16:1 n-7	2.20 ± 0.95 ^c^	3.23 ± 0.37 ^b^	4.86 ± 1.48 ^a^	0.001
C17:0	0.91 ± 0.20 ^a^	0.70 ± 0.22 ^b^	0.72 ± 0.11 ^ab^	0.027
C18:0	19.0 ± 2.3	18.0 ± 2.0	17.3 ± 1.9	0.195
C18:1 n-9	14.6 ± 2.0 ^b^	20.1 ± 1.9 ^a^	19.8 ± 4.7 ^a^	<0.001
C18:2 n-6	21.0 ± 3.3 ^a^	9.06 ± 1.19 ^c^	10.5 ± 0.9 ^b^	<0.001
C18:3 n-3	n.d. ^b^	0.82 ± 0.17 ^a^	1.00 ± 0.13 ^a^	0.022
C18:3 n-6	0.66 ± 0.15 ^a^	0.26 ± 0.05 ^c^	0.36 ± 0.06 ^b^	<0.001
C20:2 n-6	0.65 ± 0.23 ^a^	0.37 ± 0.09 ^b^	0.22 ± 0.07 ^c^	<0.001
C20:3 n-6	1.03 ± 0.76	1.13 ± 0.26	1.17 ± 0.20	0.825
C20:4 n-6	18.5 ± 4.9	15.8 ± 3.9	15.9 ± 3.1	0.255
C22:5 n-3	0.54 ± 0.20 ^a^	0.22 ± 0.08 ^b^	0.29 ± 0.08 ^b^	<0.001
C22:6 n-3	3.44 ± 1.30	2.76 ± 1.30	2.98 ± 1.22	0.483
Total	112 ± 11	103 ± 11	107 ± 12	0.200

Data are means ± SD for *n* = 10 rats/group. Means not sharing the same superscript letter (^a, b, c^) differ across groups, *p* < 0.05.

**Table 4 animals-13-03356-t004:** Fatty acid concentrations in the *M. gastrocnemius* of lean Zucker rats fed semi-synthetic diets with either soybean oil (SO), palm oil (PO), or *Hermetia* fat (HF) as the main fat source for 4 weeks.

	SO	PO	HF	*p*-Value
*Fatty acid, µmol/g*				
C12:0	0.48 ± 0.16 ^b^	2.34 ± 1.53 ^a^	1.68 ± 1.52 ^a^	<0.001
C14:0	0.65 ± 0.32	1.12 ± 0.65	1.07 ± 0.83	0.247
C14:1 n-5	0.19 ± 0.13	0.22 ± 0.03	0.22 ± 0.16	0.249
C16:0	13.9 ± 4.7	13.4 ± 3.9	14.8 ± 7.4	0.855
C16:1 n-7	1.28 ± 0.65	1.39 ± 0.82	1.75 ± 1.52	0.756
C17:0	0.23 ± 0.11	0.15 ± 0.05	0.20 ± 0.05	0.139
C18:0	6.01 ± 1.61	4.81 ± 0.80	5.37 ± 0.91	0.145
C18:1 n-9	9.97 ± 5.53	8.42 ± 3.80	10.7 ± 7.2	0.690
C18:2 n-6	16.7 ± 8.3 ^a^	8.58 ± 2.58 ^b^	9.23 ± 4.01 ^b^	0.016
C18:3 n-3	0.83 ± 0.54 ^a^	0.23 ± 0.15 ^b^	0.33 ± 0.27 ^b^	0.008
C20:4 n-6	4.43 ± 1.02	3.46 ± 0.54	4.39 ± 1.08	0.056
C22:5 n-3	0.55 ± 0.13 ^a^	0.35 ± 0.07 ^b^	0.40 ± 0.14 ^ab^	0.004
C22:6 n-3	1.76 ± 0.72 ^a^	1.05 ± 0.34 ^b^	1.56 ± 0.47 ^ab^	0.028
Total	56.9 ± 23.2	54.4 ± 14.6	51.6 ± 25.1	0.530

Data are means ± SD for *n* = 10 rats/group. Means not sharing the same superscript letter (^a, b^) differ across groups, *p* < 0.05.

## Data Availability

All data are presented in the manuscript and its Appendix A.

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
