# Peer review of "Effect of Hermetia illucens Fat, Compared with That of Soybean Oil and Palm Oil, on Hepatic Lipid Metabolism and Plasma Metabolome in Healthy Rats"

_animals, 2023, doi:10.3390/ani13213356_

Round 1

Reviewer 1 Report

Comments and Suggestions for Authors

The present study evaluated the effect of an insect derived oil (HF) compared with palm Oil and Soy Oil on the hepatic lipid metabolism and the plasma metabolome of healthy rats. The main finding of the present study is that HF exerts no liver and plasma lipid modulating effects as compared to PO and SO in healthy rats.  Considering the lack of studies investigating the metabolic effects of HF, the evidence of this study appear relevant when considering alternative fat source for food production.

Minor comments

Line 152-158 This part should be included in the statistical analysis paragraph 2.9.

Table 1 The line concerning the initial body weight could be deleted and included within the text in line 72-73 considering the standardized origin of the experimental animals (7-8-wk of age homozygous (fa/fa) lean Zucker rats (Crl:ZUC-Leprfa/+- Charles River (Sulzfeld, Germany)).   

The specification of right organs sampling should be indicated in the material and methods and omitted from the table

Author Response

Reviewer 1

The present study evaluated the effect of an insect derived oil (HF) compared with palm Oil and Soy Oil on the hepatic lipid metabolism and the plasma metabolome of healthy rats. The main finding of the present study is that HF exerts no liver and plasma lipid modulating effects as compared to PO and SO in healthy rats. Considering the lack of studies investigating the metabolic effects of HF, the evidence of this study appears relevant when considering alternative fat source for food production.

Minor comments

  1. Line 152-158 This part should be included in the statistical analysis paragraph 2.9.

    Our response: We followed this Please see lines 173-179.
  2. Table 1 The line concerning the initial body weight could be deleted and included within the text in line 72-73 considering the standardized origin of the experimental animals (7-8-wk of age homozygous (fa/fa) lean Zucker rats (Crl:ZUC-Leprfa/+- CharlesRiver (Sulzfeld, Germany)).

    Our response: We followed this suggestion. Please see lines 78-79.
  1. The specification of right organs sampling should be indicated in the material and methods and omitted from the table.

    Our response: We followed this suggestion. Please see lines 114-115.

Reviewer 2 Report

Comments and Suggestions for Authors

In this article, the authors investigated the effect of HF compared to PO and SO on healthy rats' hepatic lipid metabolism and plasma metabolome. Their results showed that HF exerted no adverse metabolic outcomes in healthy rats compared to PO or SO and suggested that HF could be a safe alternative fat source to PO for food production. This interesting research tried to look for a safe alternative fat source that may be more environmentally friendly for food production industries. The overall structure is good quality, and the methods and results are straightforward to readers. Comments are listed below.

1. In Figure 1B, the images are not clear. Please provide more clear images.  

2. Please explain more clearly the loading plot in Figure 4B. What are the red spots indicated? What is the meaning of this plot?

3. The authors discussed that a high intake of SFA is considered to impair cardiovascular health. In this study, the fat contains only counts for about 7% of the daily diet. The fat proportion in the diet is relatively low and the feeding time (4 weeks) is too short to cause adverse outcomes. Thus, the results are unsustainable to support the conclusion that HF is a safe alternative fat source to PO for food production. It still needs more comprehensive studies to determine whether a high intake of HF could cause any adverse health outcomes. 

Comments on the Quality of English Language

Please check carefully that some errors need to be corrected. 

Author Response

Reviewer 2

In this article, the authors investigated the effect of HF compared to PO and SO on healthy rats' hepatic lipid metabolism and plasma metabolome. Their results showed that HF exerted no adverse metabolic outcomes in healthy rats compared to PO or SO and suggested that HF could be a safe alternative fat source to PO for food production. This interesting research tried to look for a safe alternative fat source that may be more environmentally friendly for food production industries. The overall structure is good quality, and the methods and results are straightforward to readers. Comments are listed below.

  1. In Figure 1B, the images are not Please provide more clear images.

    Our response: We agree with this view. Thus, we increased resolution of the images to improve clarity.

  2. Please explain more clearly the loading plot in Figure 4B. What are the red spots indicated? What is the meaning of this plot?

    Our response: Thanks for this comment. In order to improve clarity, we added the following to the description of loading plot: “The loading plot provides insight into the parameters that drive the separation of the data along the principal component. The red dots in the loading plot mark the ten parameters in each direction (top, bottom, left and right) that contribute most to the separation of the data points along the principal ” (lines 289-292).

  3. The authors discussed that a high intake of SFA is considered to impair cardiovascular In this study, the fat contains only counts for about 7% of the daily diet. The fat proportion in the diet is relatively low and the feeding time (4 weeks) is too short to cause adverse outcomes. Thus, the results are unsustainable to support the conclusion that HF is a safe alternative fat source to PO for food production. It still needs more comprehensive studies to determine whether a high intake of HF could cause any adverse health outcomes.

    Our response: We appreciate this comment very much. Thus, we added the following to the discussion section: “Despite that no adverse effects of HF were found in the present study, firm conclusions about the safety of HF cannot be drawn from this study due to the rather short feeding period. In addition, the transfer of the data to the human situation is limited by the fact that the fat content of the diets was comparatively low. While the rat diets of the present study contained 16% of calories from fat, typical human diets in westernized countries contain 35-40% of calories from fat.” (lines 372-377). In addition, we slightly modified the conclusion section: “Thus, the present findings indicate that HF is a potential alternative fat source to PO for food production. Long-term feeding studies and studies with higher dietary fat lev-els are required to address safety issues associated with HF intake.” (lines 383- 386).

  4. Please check carefully that some errors need to be corrected.

    Our response: We followed this suggestion, and thoroughly checked the manuscript for any errors.

Reviewer 3 Report

Comments and Suggestions for Authors

The manuscript describes the effects of replacing fat from palm and soybean oil with fat from Hermetia insects. The use of insects in feeding farm animals is currently of great interest due to low production costs and relatively low environmental impact. However, before this type of feed becomes standardly used in agriculture, it is necessary to check whether it does not cause any adverse effects on the functioning of the body. The authors of the manuscript did not find any negative effects of the use of fat from insects, on the liver and plasma lipid concentration or on SCFA concentartion in cecum digesta. however, they identified some changes in the plasma metabolome. The manuscript is well written and the data are clearly presented, however I suggest to add some information about previous studies on Hermetia feeding effect on animal physiology in the discussion.

More detailed comments

line15 -illucens - please change into italics

line 19 - which were fed?

line 20 wk - week

line 32 performance data?

Introduction:

Please provide some refernces that farming insects is enviromentaly friendly in comparison to other feedstuff production

Table 1 please provide the energy data for each diet, were the diets isocaloric? SFA/ UFA ratio and N6/N3 ratio of the diets

M&M

line 161 - microbiome data?

line 197 Were the fatty acid content in the feedstuff correlated with the fatty acid content in the liver and muscle. Why fatty acid content was not analysed in the adipose tissue?

line 241 den?

line 242 plaese explain C2, C3...

line 300 No plasma lipid modulating effect? what about metabolome results?

line 320-321 are the groups correct? ther are higher difference between SO & HF than between PO&HF according to PCA

line 345 It is generally accepted that nutrition may modulate microbiome even in healthy organism

Discussion should include some previous studies on hermetia insects effect on animal physiology

Author Response

Reviewer 3

The manuscript describes the effects of replacing fat from palm and soybean oil with fat from Hermetia insects. The use of insects in feeding farm animals is currently of great interest due to low production costs and relatively low environmental impact. However, before this type of feed becomes standardly used in agriculture, it is necessary to check whether it does not cause any adverse effects on the functioning of the body. The authors of the manuscript did not find any negative effects of the use of fat from insects, on the liver and plasma lipid concentration or on SCFA concentration in cecum digesta. however, they identified some changes in the plasma metabolome. The manuscript is well written and the data are clearly presented, however I suggest to add some information about previous studies on Hermetia feeding effect on animal physiology in the discussion.

More detailed comments

  1. line15 -illucens - please change into
    Our response: We followed this suggestion. Please see lines 15-16.

  2. line 19 - which were fed?
    Our response: We added the missing “fed” to this Please see line 19.
  1. line 20 wk – week.
    Our response: We followed this suggestion and changed wk into weeks in the simple Please see line 20.
  2. line 32 performance data?
    Our response: We changed performance data to “Feedintake and body weight gain”. Please see lines 32-33.
  1. Introduction: Please provide some references that farming insects is environmentally friendly in comparison to other feedstuff production.
    Our response: We followed this suggestion and added and commented two Please see lines 52-58 and refs. 5 and 6.
  2. Table 1 please provide the energy data for each diet, were the diets isocaloric? SFA/ UFA ratio and N6/N3 ratio of the diets
    Our response: We followed this suggestion and added the SFA/UFA ratio and the n-6/n-3 ratio to table 1. We did not analyze the gross energy content of the diets, because the three diets contained the identical basal diet (all diet components except the fats). Considering that the energy contents of soybean oil, palm oil and Hermetia illucens fat are comparable, the three diets are roughly isocaloric.
  3. M&M: line 161 - microbiome data?
    Our response: Thank you for critically reading our This information was a mistake. We deleted this.
  4. line 197 Were the fatty acid content in the feedstuff correlated with the fatty acid content in the liver and Why fatty acid content was not analysed in the adipose tissue?
    Our response: Thanks for these interesting questions. As described in section 3.3 and 3.4, fatty acid concentrations in liver and skeletal muscle total lipids were influenced by the dietary fats. For instance, the concentrations of C12:0, C14:0 and C16:1 n-7, whose levels were highest in the HF fat, were higher in the liver of the HF group than in the other groups (Table 3), indicating that liver fatty acid composition reflected the dietary fatty acid composition (please see lines 209-218). In the skeletal muscle, this dependence was less clear, but the concentration of C12:0 – the dominating fatty acid in HF fat – was also higher in the HF group than in the SO group (lines 225-231). Because analysis of liver and skeletal muscle fatty acid composition showed the expected effect of the dietary fats on tissue fatty acid composition, additional analysis of adipose tissue fatty acid composition was not necessary.
  1. line 241 den?
    Our response: Thanks for thoroughly reading the We corrected this mistake (line 252).
  2. line 242 please explain C2, ..
    Our response: We followed this suggestion. Please see lines 253-254.
  3. line 300 No plasma lipid modulating effect? what about metabolome results?
    Our response: No lipid modulating effects referred to TG and Chol levels, which were not affected by the type of fat. To improve clarity we changed “lipid-modulating” to “TG- and Chol-modulating”. Please see lines 316-317.
  4. line 320-321 are the groups correct? there are higher difference between SO & HF than between PO&HF according to
    Our response: We agree with the reviewer, that the difference between SO and HF and PO was stronger that between HF and PO. Thus, we corrected this mistake to “more pronounced differences”. Please see line 336.
  5. line 345 It is generally accepted that nutrition may modulate microbiome even in healthy
    Our response: We agree with the reviewer and deleted this statement.
  6. Discussion should include some previous studies on hermetia insects effect on animal
    Our response: We followed this suggestion and added some previous studies on Hermetia insects in monogastric animals, which we discussed as follows: “While the present study does not provide an answer on the question if the fat from Hermetia illucens modulates the gut microbiome of healthy rats, a large number of studies exists demonstrating an effect of either the fat or the larvae meal from Hermetia illucens on the gut microbiome and the intermediary metabolism of monogastric animals [11,25,26].”. Please see lines 360-364 and refs. 25 and 26.

Round 2

Reviewer 2 Report

Comments and Suggestions for Authors

The authors have appropriately responded to the comments. I have no further comment. 

Reviewer 3 Report

Comments and Suggestions for Authors

All my comments have been addressed